# Cardiotoxicity, Cardioprotection, and Prognosis in Survivors of Anticancer Treatment Undergoing Cardiac Surgery: Unmet Needs

**DOI:** 10.3390/cancers15082224

**Published:** 2023-04-10

**Authors:** Vittorio Palmieri, Maria Teresa Vietri, Andrea Montalto, Andrea Montisci, Francesco Donatelli, Enrico Coscioni, Claudio Napoli

**Affiliations:** 1Unit of Cardiac Surgery, Cardiovascular Department, Azienda Ospedaliera di Rilevanza Nazionale “San Sebastiano e Sant’Anna”, 81100 Caserta, Italy; 2Department of Precision Medicine, School of Medicine, “Luigi Vanvitelli” University of Campania, 80100 Naples, Italy; 3Division of Cardiothoracic Intensive Care, Cardiothoracic Department, ASST Spedali Civili, 25123 Brescia, Italy; 4Department of Cardiac Surgery, Istituto Clinico Sant’Ambrogio, 20161 Milan, Italy; 5Cardiac Surgery, University of Milan, 20122 Milan, Italy; 6Division of Cardiac Surgery, AOU San Giovanni di Dio e Ruggi D’Aragona, 84131 Salerno, Italy; 7Department of Advanced Medical and Surgical Sciences (DAMSS), “Luigi Vanvitelli” University of Campania School of Medicine, 80100 Naples, Italy

**Keywords:** cancer, cardiac surgery, prognosis

## Abstract

**Simple Summary:**

Cardiovascular (CV) risk factors and disease are increasingly reported among survivors of anticancer treatments as those treatments are effective in improving prognosis in patients affected by malignancy. It is unclear the extent to which drugs used in general population to prevent ischemic heart disease, valvular heart disease and heart failure, and aortic syndromes, which are related to CV risk factors and preclinical CV disease, are effective in survivors of anticancer treatments. Among those who survived to anticancer treatments, prognosis after cardiac surgery treatments may differ from that in the general population, and may require specific pre-surgery risk assessment.

**Abstract:**

Background: Anticancer treatments are improving the prognosis of patients fighting cancer. However, anticancer treatments may also increase the cardiovascular (CV) risk by increasing metabolic disorders. Atherosclerosis and atherothrombosis related to anticancer treatments may lead to ischemic heart disease (IHD), while direct cardiac toxicity may induce non-ischemic heart disease. Moreover, valvular heart disease (VHD), aortic syndromes (AoS), and advanced heart failure (HF) associated with CV risk factors and preclinical CV disease as well as with chronic inflammation and endothelial dysfunction may also occur in survivors of anti-carcer treatments. Methods: Public electronic libraries have been searched systematically looking at cardiotoxicity, cardioprotection, CV risk and disease, and prognosis after cardiac surgery in survivors of anticancer treatments. Results: CV risk factors and disease may not be infrequent among survivors of anticancer treatments. As cardiotoxicity of established anticancer treatments has been investigated and is frequently irreversible, cardiotoxicity associated with novel treatments appears to be more frequently reversible, but also potentially synergic. Small reports suggest that drugs preventing HF in the general population may be effective also among survivors of anticancer treatments, so that CV risk factors and disease, and chronic inflammation, may lead to indication to cardiac surgery in survivors of anticancer treatments. There is a lack of substantial data on whether current risk scores are efficient to predict prognosis after cardiac surgery in survivors of anticancer treatments, and to guide tailored decision-making. IHD is the most common condition requiring cardiac surgery among survivors of anticancer treatments. Primary VHD is mostly related to a history of radiation therapy. No specific reports exist on AoS in survivors of anticancer treatments. Conclusions: It is unclear whether interventions to dominate cancer- and anticancer treatment-related metabolic syndromes, chronic inflammation, and endothelial dysfunction, leading to IHD, nonIHD, VHD, HF, and AoS, are as effective in survivors of anticancer treatments as in the general population. When CV diseases require cardiac surgery, survivors of anticancer treatments may be a population at specifically elevated risk, rather than affected by a specific risk factor.

## 1. Introduction

Cardiovascular (CV) toxicity of anticancer treatments may become the second most common cause of heart failure (HF) world-wide [1]. Several mechanisms may explain the relationship between anticancer treatments and myocardial disease potentially leading to HF [2,3], including ischemic heart disease (IHD), the most common cause of HF in the general population (Table 1). It should be, in fact, considered that CV risk factors are common among survivors of anticancer treatments [3,4] while endothelial dysfunction related to metabolic syndrome and chronic systemic inflammation are likely associated with cancer and anticancer treatments, and may induce atherosclerosis and trigger atherothrombosis [5,6,7]. In fact, as expected, pre-existing CV risk factors and diseases increase the likelihood of CV events in survivors of anticancer treatments [8,9,10,11,12,13]. Among survivors of anticancer treatments, the need for prognostically relevant myocardial revascularization and treatment of valvular heart disease (VHD) may be more than generally expected [14]. To date, relevant cardiac events are disclosed only in part in trials focusing on anticancer treatments [15]. Moreover, only small reports are available on the frequency and prognosis of aortic disease or aortic syndromes (AoS) among survivors of anticancer treatments [16]. Largely due to refined anticancer therapy, and multidisciplinary collaboration [17], the issue is rising in importance in terms of epidemiology in the last decade. In fact, as survival in cancer diseases increases [18], we may likely face an increasing number of patients with CV disease requiring cardiac surgery among those with a history of anticancer treatments. Among those, it remains to be established the extent to which indications to cardiac surgery for coronary artery disease (CAD), VHD, advance HF, or AoS by established guidelines [19,20,21,22,23] hold among patients with a history of anticancer treatments compared to the general population. Moreover, whether previous anticancer treatments should be a risk factor or, actually, indicate a population at a specifically elevated risk of untoward events after cardiac surgery is unclear as much as relevant. Indeed, the prognosis impact of cardiac surgery has been estimated mostly in the general population, has elevated personal biological cost, and impacted quality of life in the short and medium term through potential worsening of pre-surgery conditions such as renal insufficiency, anemia, infection disease, chronic lung disease, and mobility and frailty. Those conditions may well be present and dominate the clinical outlook of survivors of anticancer treatments, impacting indications.

## 2. Methods

Electronic libraries have been searched by standard methodology [24], using the following terms: “cancer”, “treatment”, “survivor”, “cardiac”, “surgery”, and “prognosis”. Original manuscripts published in the last two years in English in peer-reviewed journals have been overviewed and considered. Furthermore, ongoing trials have been identified using clinicaltrials.gov (https://clinicaltrials.gov/ (accessed on 1 February 2023)) and EudraCT (https://eudract.ema.europa.eu/ (accessed on 1 February 2023)) employing the terms: “cardiac toxicity”, “trastuzumab”, “cancer”, “cardiac”, “cardiovascular cancer”, “immune checkpoint inhibitors”, “CAR-T therapy”, “cardiovascular diseases”, and “heart diseases”. No specific bias or study quality assessments, or additional inclusion/exclusion criteria, were employed.

## 3. Results

### 3.1. Anticancer Treatments, Cardiac Toxicity, and Cardioprotection as of Today

As reported in Table 1, and in more detail elsewhere [3], the majority of anticancer treatments are associated with non-IHD reduction of LV ejection fraction (EF), while a minority show primary VHD. No reports exist on a specific association between anticancer treatments and AoS, while reports on aortic disease in survivors of anticancer treatments remain limited or anecdotical [16].

#### 3.1.1. Anthracycline

Anthracycline antineoplastic agents (AC), such as doxorubicin, are effective components and widely used in adjuvant chemotherapy against breast cancer, lymphomas, leukemias, and sarcomas. The model of directly induced myocardial apoptosis has been described with AC as a predominantly non-inflammatory non-atherothrombotic myocardial damage predisposing to HF. Myocardial damage caused by AC have been documented by several experimental studies, and recognized post-marketing clinical practice. Non-IHD characterized by systolic dysfunction may progress to overt HF, and untoward nonfatal and fatal events [25].

AC-related cardiotoxicity is cumulative-dose-dependent. It is well-established that oxidative stress contributes to cardiomyocyte dysfunction and death related to the use of anthracyclines. Mitochondrial dysfunction is responsible for the large part of the oxidative stress, as documented previously [26].

Careful and aggressive monitoring of cardiac function is needed in patients treated with AC to prevent major CV events [27]. In a study conducted in 120 patients, cardiac function recovery after anthracycline-induced LVEF reduction was reported in less than 10% of the treated [28]. LV EF recovery, partial up to normal, may increase significantly by the use of angiotensin I converting enzyme inhibitors (ACEi) and/or carvedilol [29], especially when introduced within 6 months from the anticancer treatments [30].

#### 3.1.2. Trastuzumab

As cardiotoxicity is a relatively frequent effect of trastuzumab by a mechanism not yet fully understood, effective cardioprotective interventions in persons treated with trastuzumab remain largely unclear and unexplored [31]. In patients treated with trastuzumab, LVEF may decline, triggering the withholding of trastuzumab for 4 weeks or more, and a safe restart of the medication if the LVEF returns to normal or to a value close to that pre-treatment. The more recent recommendation is to continue trastuzumab therapy in association with general cardioprotective therapy rather than withhold trastuzumab [32]. At variance to ACs, the trastuzumab-related cardiotoxicity is not dose-dependent, is reversible, and does not occur systematically. Yet, trastuzumab is often associated with ACs for anticancer treatments. There are three main mechanisms leading to cardiac toxicity in persons receiving trastuzumab: (1) direct damage of the myocardium, with the clinical expression of LV dysfunction, associated with a histologic signs of myocardial edema and inflammation [33]; (2) pro-arrhythmic effects leading to the development of brady-arrhythmias (sick sinus node syndrome) and tachyarrhythmias (incessant high frequency atrial flutter and/or fibrillation and tachy-cardiomyopathy) [34]; (3) development of insulin resistant/metabolic syndrome, impaired glucose tolerance, abdominal obesity, reduced high-density lipoprotein (HDL) cholesterol levels, elevated triglycerides, and hypertension [3]. Indeed, it has been observed that breast cancer patients with metabolic syndrome developed after or enhanced by trastuzumab therapy suffered from subclinical myocardial dysfunction [35,36,37]. ACEi or carvedilol reduced the rate of trastuzumab-related cardiotoxicity in HER2-positive breast cancer patients treated for 12 months [38]. In another study in 222 patients with HER-2 positive breast cancer, it was observed that low baseline LVEF and greater LVEF decline post-anthracycline were both independent predictors of trastuzumab-related cardiotoxicity [39]. A phase two study evaluating folinic acid, fluorouracil, and oxaliplatin (FOLFOX) plus trastuzumab as second- or third-line treatment for HER-2 positive biliary tract carcinoma observed no treatment-related cardiac toxic effects or deaths, while the overall quality of health assessment score (EuroQoL-VAS) did not change significantly throughout the treatment [40]. Moreover, a multicenter randomized phase II trial evaluated the safety, tolerability, and toxicity profile of trastuzumab in patients with HER2 advanced or recurrent endometrial carcinoma, and observed that trastuzumab appears to be safe and has a manageable toxicity profile both when used in combination with chemotherapy and when used as monotherapy [41].

When trastuzumab is used in combination with ACs, the cardiotoxic effects of both drugs may be synergic. It was shown that the prevalence of New York Heart Association classification (NYHA) grade III or IV HF may reach 16% among persons receiving the two medications in combination, and cyclophosphamide. Of those, 8% developed symptomatic cardiac dysfunction and withheld trastuzumab [42], impacting unfavorably the cancer-related prognosis.

#### 3.1.3. Immune Checkpoint Inhibitors (CTLA-4 Blockers and PD1/PDL1 Blockers)

Over the past 10 years, novel immune-based anticancer therapies known as immune checkpoint inhibitors (ICIs) have been introduced. Some concern exists on potential CV toxicity with ICI therapy in detail [43]. As ICI is more frequently prescribed, the ongoing reporting and prevalence of diagnosis-related CV toxicity changes, so useful data and recommendations evolve rapidly [44,45]. ICIs are relevant in a variety of cancer diseases, such as melanoma, non-small cell lung cancer (NSCLC), hepatocellular carcinoma (HCC), renal cell carcinoma (RCC), and Hodgkin lymphoma. ICIs act upon T lymphocytes and antigen-presenting cells, versus programmed cell death protein 1 (PD1), programmed cell death protein ligand 1 (PD-L1), and cytotoxic T-lymphocyte antigen 4 (CTLA-4), deleting the immune tolerance of the T cells against cancer cells [46]. ICIs may be associated with pericarditis, arrhythmias, cardiomyopathy, and acute coronary syndrome; in particular, myocarditis has been found in 0.1% of the patients taking nivolumab as monotherapy, and 0.3% in those treated with nivolumab and ipilimumab [47], with a reported mortality rate in systematic reviews as high as 50% [48]. Recently, pembrolizumab (anti PD-1) used in 30 patients affected by classic and endemic Kaposi’s sarcoma showed effective anti-tumor activity with an acceptable safety profile, as cardiac treatment-related adverse events occurred in 6% (acute HF) and discontinuation was required in two (12%, due to acute reversible HF) [49].

#### 3.1.4. CAR-T Therapy

Anticancer treatment based on chimeric antigen receptor T-cell (CAR-T) is associated with a significant improvement of the prognosis in several hematologic malignancies, and is therefore used with increased frequency. However, the more the treatment is used, the more reports highlight some cardiotoxicity effects, potentially related to cytokine release syndrome (CRS), and inflammatory activation is sustained by circulating cytokines, including myocardial injury, tachyarrhythmias, ventricular arrhythmias and atrial fibrillation, myocardial ischemia, and venous thromboembolism [50,51]. CV ultrasound imaging surveillance of subjects treated with CAR-T identified earlier and prevented clinically overt severe cardiotoxicity and cardiovascular complications [52], as cardiac toxicity related to CAR T-cell-associated CRS may be resolved spontaneously at day-28 post infusion [53].

### 3.2. Anticancer Treatments, CV Risk and Disease, and Cardiometabolic Protection

As CV disease may be frequent among survivors of anticancer treatments, overall large and nonspecific trials reported a relatively low prevalence of patients with histories of anticancer treatments (2–4% as reported above), potentially because of selection biases. As reported in Table 2, a number of studies are ongoing on the specific issue of cardiac protection in anticancer treatments, because cardiac disease in survivors of anticancer treatments emerge as an additional and relevant prognosticator.

### 3.3. Cardiovascular Disease, Cardiac Surgery and Prognosis in Survivors of Anticancer Therapy

It remains unclear whether subjects with a history of anticancer treatments may be considered a population extracted from a more general population of subjects with CV risk factors, or a population with a specifically increased CV risk leading to CV disease. Moreover, CV disease may lead to indication to cardiac surgery in persons with a history of anticancer therapy. In survivors of anticancer treatments, specific estimates of prognosis and impact on quality of life in the mid and long terms are key factors to establish whether those procedures are appropriate, beyond the exception of immediately life-threating conditions. When it comes to estimating the risk of cardiac surgery, it emerges that the Euroscore II [61] does not account for previous anticancer treatment while the Society of Thoracic Surgeons score (the STS score) [62] weights both the history of mediastinal radiation and the history of cancer within 5 years. Both Euroscore II score and STS score estimate the risk of mortality in the short term. Nevertheless, no risk score to establish prognosis and impact on quality of life after cardiac surgery have been developed specifically in survivors of anticancer treatments. In historical series, an unfavorable bias may likely have reduced the prevalence of patients with histories of anticancer treatments among the candidates to cardiac surgery, preventing the development of specific algorithms to estimate prognoses related to cardiac surgery in such a specific population. Hence, several unmet needs on the matter of the transition from CV risk factors to CV disease, and to establish prognosis after cardiac surgery in survivors of anticancer treatments deserve attention: (1) whether specific therapeutic targets for CV prevention should be established in survivors of anticancer treatments; (2) the extent to which CV prognosis may compete with that related to cancer among survivors of anticancer treatments at a certain point in the follow-up; (3) the extent to which the history of anticancer treatments impact CV prognosis independently among those undergoing cardiac surgery; and (4) the extent to which extracorporeal circulation (ECC) during cardiac surgery impacts CV and global prognosis in survivors of anticancer treatments. Therefore, we reviewed prognoses in cardiac surgery patients presenting with CAD, VHD, or AoS exposed to anticancer treatments with potential cardiac toxicity, in order to account for the extent to which the Literature may be solid in supporting surgical treatments in such a population of patients.

The prevalence of CV risk factors and of overt CV disease among survivors of anticancer treatments may be higher than in the general population, requiring specific attention and aggressive therapeutic targets to prevent overt CV disease [3]. Indeed, survivors of anticancer treatments might be considered subjects with preclinical CV disease [3], which justifies aggressive targets for CV prevention [63]. Hence, in survivors of anticancer treatments, preclinical CV disease should be systematically searched for, taking into account CV risk factors and evident CV diseases identified before anticancer treatments. Ultrasound and non-ultrasound imaging-based modalities, as well as measurement of circulating biomarkers (cardiac troponins and natriuretic peptides), may help to identify preclinical CV disease, or to make diagnosis of symptomatic CV disease [3,63].

In a series of 4474 cases collected in 8 years [13], the history of cancer disease and previous anticancer treatments was as high as 2%, comprising mostly (2/3 of the patients, approximately) men with a history of solid cancer; the patients who underwent cardiac surgery showed a prognosis at 30-day follow-up comparable to matched controls. However, patients with a history of malignancy had an increased rate of complications such as blood transfusion, re-intubation, ventilator-related pneumonia, septicemia, and coagulation abnormalities impacting the length of their stay in the intensive care unit and the hospital overall. The former study did not investigate the prognosis at longer follow-up and impact of cardiac surgery on quality of life. In a larger series of 8620 patients referred for cardiac surgery [64], 2% had histories of cancer disease, and those patients had pre-surgery CV treatments comparable to controls; long-term post-cardiac surgery survival was 23% lower in survivors of anticancer treatments than in controls, essentially predicted by pre-surgery LV EF lower in survivors of anticancer treatments than in controls, and shorter time between diagnosis of cancer and cardiac surgery. In retrospective studies including patients who underwent cardiac surgery, a minority had previous diagnoses and treatments for cancer diseases, and their mortality rates at year-1 follow-ups were higher in those exposed to anticancer treatments as compared to controls, without reaching statistical significance. However, progression in cancer disease was the most common cause of death among those patients who underwent cardiac surgery [64,65,66]. In a single-center experience of over 20 years of lung cancer, those who underwent elective cardiac surgery for IHD showed a relatively low rate of death in the short-term (3%), and the 5 years overall mortality was 38% while the 5 years disease-free survival was 59%; factors predicting death after thoracic surgery were post-surgery blood transfusions and multiple organ failure, while post-cardiac surgery cancer recurrence was predicted by pre-surgery cancer stage and shorter lag between cardiac and lung procedures [67]. In those studies, no specific information is reported on a comparison between predicted and observed mortality or impact on quality of life post cardiac surgery.

A history of chest radiation increased the risk of surgical complications, well predicted by preoperative computed tomography and echocardiography imaging [68,69]. Radiation impacts thoracic structures including vascular structures, valves, pericardium/myocardium, and the conduction system, as well as the autonomic system [70]. Radiation-related aortic calcification and internal mammary artery fibrotic degeneration may preclude grafting specific coronary artery bypass (CABG), impacting surgical strategies and post-surgical perspectives and prognoses. Thick, fibrotic, and retracted valvular leaflets, valvular calcification, degenerative regurgitation, and progressive stenosis may be common within the first two decades post radiation therapy. [70] Left VHD predominates, and stenosis or regurgitation of the aortic valve may be as frequent as 1% within year-10 follow-up, 4% within year-15 follow-up, and 6% at and 9% by year-20 and year-25 follow-ups, respectively. [70] Rarely, cardiac valves are normal after chest radiation therapy [71].

Furthermore, there is no evidence that ECC impacts cancer-related prognosis in survivors of anticancer treatments [72,73].

Overall, in survivors of anticancer therapy undergoing cardiac surgery, a specific assessment of CV risk profile [3] predictors of peri-procedural untoward events is specifically recommended [14].

### 3.4. Advanced HF in Survivors of Anticancer Treatments: Updates to the Surgical Options

Patients with previous histories of cancer and anticancer treatments may develop advanced HF [74], which may require surgical treatments such as implantation of a mechanical LV assist device (LVAD) bridge to candidacy, transplantation or destination, total artificial heart (THA) as bridge to transplantation, and finally heart transplantation [75]. Heart transplantation may be executed in survivors of anticancer treatment. However, in order to include patients in the list of subjects awaiting heart transplantation, survivors of anticancer treatments must be free from recurrence of cancer for a lag time which may vary among centers [76]. Of note, survivors of anticancer treatments may show a specific profile of frailty, which is a relevant predictor of untoward events after LVAD implantation as well as after heart transplantation [77,78,79]. In fact, in survivors of anticancer treatments, post-heart transplantation prognosis is proportional to the length of the period free of cancer disease prior to heart transplantation [80,81]. LVAD implantation appears to be a reasonable option in patients with advanced HF and concomitant history of malignancy, as overall survival in the mid-term may be as high as 50% in such a group of patients [82]; importantly, events post LVAD implantation rarely included reactivation of cancer, but most frequently included LVAD-related events impacting quality of life and survival expectancy, which requires a very careful and tailored evaluation in single patients. Experience of THA in patients with amyloidosis used as bridge to heart transplantation appears anecdotical [83], so far, while we face increased need for treatments among patients with cardiac amyloidosis who may receive specific treatments limited to cardiac dysfunction as bridge to heart transplantation [84].

## 4. Conclusions

Current anticancer therapies are associated with improved survival in patients fighting cancer. However, cancer and anticancer treatments may induce direct cardiac toxicity, chronic inflammation, vasculitis and coronary disease, metabolic syndromes, and endothelial dysfunction leading to atherosclerosis and atherothrombosis. IHD, nonIHD, VHD, HF, and AoS are the common CV phenotypes of CV disease in survivors of anticancer treatments. Specific and systematic experience are lacking in the extent to which current medications used in the general population to prevent and treat CV disease, protect from the decline in LV EF, and prevent HF are effective in subjects that survived anticancer treatments. Yet, general agreement exists on the fact that CV preventions should be set on aggressive targets in survivors of anticancer treatments, which intrinsically consider such a group of patients with a specifically elevated risk of events rather than patients with a specific additional risk factor. Moreover, CV diseases, HF, and AoS among survivors of anticancer treatments may require programmed cardiac surgery. However, it remains to be evaluated in depth whether previous anticancer treatment should be an indicator of risk or if it should identify a population at specifically elevated risk when undergoing cardiac surgery. In fact, historically, anticancer therapy may have acted as a negative bias in clinically oriented series, registries, and spontaneous observations exploring prognosis post cardiac surgery. However, as prognosis in cancer diseases improves by specific treatments, there is an increased need to develop specific models predicting mortality, morbidity, and quality of life in the mid and long term in survivors of anticancer treatments, in order to establish whether cardiac surgery, including LVAD, TAH, and heart transplantation, are sustainable therapeutic options in such a group of patients.

## Figures and Tables

**Table 1 cancers-15-02224-t001:** Cardiovascular phenotypes associated with anticancer treatments (reported in details elsewhere [3]).

Cardiovascular Phenotype	Treatments
IHD—LV dysfunction/HF	***Alkylating agents:*** Cisplatin.
	***Antimetabolites:*** Fluorouracil; Capecitabine; Fludarabine.
	***Antimicrotubule agents:*** Paclitaxel; Vinblastine;
	***Monoclonal antibodies HRR2:*** Bevacizumab;
	***Small-molecule TKIs:*** Ponatinib; Sorafenib; Nilotinib; Regorafenib; Cetuximab; Erlotinib.
	***Others:*** Bleomycin.
Non-IHD LV dysfunction/HF	***Anthracyclines*:** Doxorubicin; Epirubicin;
	***Alkylating agents*:** Cyclophospamide; Ifosfamide, Melphalan.
	***Antimetabolites*:** Fluorouracil; Decitabine.
	***Antimicrotubule agents*:** Docetaxel.
	***Monoclonal antibodies*:** Rituximab; Ofatumumab; Alemtuzumab.
	***Monoclonal antibodies HER2*:** Pertuzumab; Trastuzumab.
	***Small-molecule TKIs*:** Dabrafenib; Dasatinib; Lepatinib; Pazopanib; Ponatinib; Sorafenib; Trametinib; Sunitinib; Axitinib; Nilotinib; Imatinib; Vandetanib.
	***Immune checkpoint inhibitors*:** Nivolumab; Ipilimumab; Pembrolizumab.
	***Protease inhibitors*:** Bortezomib; Carfilzomib.
	***Endocrine therapy, LHRH agonists*:** Goserelin; Leuprolide;
	***Antiandrogens*:** Flutamide; Bicalutamide; Nilutamide.
	***Chimeric antigen receptor (CAR) T cell therapy*:** Tisagenlecleucel; Axicabtageneciloleucel.
	***Others*:** Tretinoin
HVD, primary	Radiation therapy
AoS	Unreported

IHD: ischemic heart disease; HF: heart failure; LV: left ventricular; HVD: heart valvular disease; AoS: aortic syndromes.

**Table 2 cancers-15-02224-t002:** Ongoing Trials on anticancer treatments and cardioprotecion as 2021 and later.

Trial	Year	Malignancy	Intervention	Estimated Enrollment	Status
Chemotherapy-Free pCR-Guided Strategy with subcutaneous trastuzumab-pertuzumab and T-DM1 in HER2-positive early breast cancer (PHERGAIN-2) [54]	2020	HER2-Positive Early Breast Cancer	To assess 3-year recurrence-free interval (3y-RFI) in patients with previously untreated HER2[+] (IHC3+) node-negative early stage breast cancer. Primary safety objective To assess global health status decline rate at 1 year from start of neoadjuvant treatment.	393 participants	Recruiting
Evaluation of Heart Function in Breast Cancer Patients Using Trastuzumab [55]	2021	Breast Cancer	Breast cancer patients using trastuzumab, collected demographic and clinical information of all patients before chemotherapy, performed echocardiography (conventional echocardiography, three-dimensional spot tracking technology), and collected blood samples to detect plasma biology Markers (TnT, BNP, GDF-15, topoisomerase	30 participants	Recruiting
Investigating the Long-term Cardiac Sequelae of Trastuzumab Therapy [56]	2021	Breast Cancer HER2-positive Breast Cancer	Patients who were treated with anthracycline and adjuvant trastuzumab chemotherapy at least 5 years previously.	60 participants	Recruiting
Mechanisms, Predictors, and Social Determinants of Cardiotoxicity in Breast Cancer (CCT2) [57]	2021	Breast Cancer	Patients treated with doxorubicin (Adriamycin) for breast cancer Patients treated with trastuzumab (Herceptin) for breast cancer Patients treated with both doxorubicin (Adriamycin) and trastuzumab (Herceptin) for breast cancer	200 participants	Recruiting
Assessment of Myocardial Injury in Patients Treated With Immune Checkpoint Inhibitors (MIICI) [58]	2022	-	Patients receive Immune Checkpoint Inhibitor as per oncology protocol	40 participants	Recruiting
CAR T Cell Therapy Related Cardiovascular Outcomes (CARTCO) [59]	2021	B-cell Acute Lymphoblastic Leukemia B-cell Lymphoma Refractory B-cell Lymphoma Recurrent Primary Mediastinal Large B-cell Lymphoma (PMBCL) Diffuse Large B Cell Lymphoma	The primary outcome is a composite of detected abnormalities on biomarkers, transthoracic echocardiogram (TTE), or Cardiac magnetic resonance (CMR) following CAR T cell infusion. The secondary outcome measures include a composite of detected abnormalities of factors on cardiac biomarkers (troponin and N-terminal pro B-type natriuretic peptide), electrocardiogram (ECG) changes and acute heart failure.	150 participants	Recruiting
Radiation-induced Cardiac Toxicity After Non-small Cell Lung Cancer Radiotherapy [60]	2021	Non-small Cell Lung Cancer	Consecutive NSCLC patients treated with standard RT with curative intent with or without platinum-based CHT	100 participants	Recruiting

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
