# Peer review of "Cardiotoxicity, Cardioprotection, and Prognosis in Survivors of Anticancer Treatment Undergoing Cardiac Surgery: Unmet Needs"

_cancers, 2023, doi:10.3390/cancers15082224_

Round 1
Reviewer 1 Report
Authors should be congratulated for their work. Manuscript is well written and deals with an interesting topic. Cancer / anti-cancer treatment survivors are increasing and the outcomes when undergoing cardiac surgery is crucial to estabilish the indications and proportionality for treatments. References are updated. Table 1 and 2 adequately summarize evidences and ongoing trials. A figure (i.e. graphical abstract) might be useful for the readers to improve the clarity of the manuscript and focus on the core content.
Author Response
To Reviewer 1
We would like to thank the Reviewer very much for reading carefully our manuscript, and provide comment found very useful in revising it. With regard to the point raised:
Reviewer: “Authors should be congratulated for their work. Manuscript is well written and deals with an interesting topic. Cancer / anti-cancer treatment survivors are increasing and the outcomes when undergoing cardiac surgery is crucial to establish the indications and proportionality for treatments. References are updated. Table 1 and 2 adequately summarize evidences and ongoing trials. A figure (i.e. graphical abstract) might be useful for the readers to improve the clarity of the manuscript and focus on the core content.”
Reply: We thank the Reviewer very much for the useful comments provided. With the revised version, we added a graphical abstract.

Reviewer 2 Report
The review is overall well written and does indeed add new insights to the scholarly literature with respect to previously published reviews.
The presentation and critical interpretation of results of previous studies should be improved.
The Authors should provide their own critical thinking and make some relevant suggestions or conclusions.
Minor concerns:
The word "evidence" is always uncountable.
"treatmetns" should be treatments
Author Response
To Reviewer 2
We would like to thank the Reviewer very much for reading carefully our manuscript and provide very useful comments to revise it. With regard to the point raised:
Reviewer: “The review is overall well written and does indeed add new insights to the scholarly literature with respect to previously published reviews. The presentation and critical interpretation of results of previous studies should be improved. The Authors should provide their own critical thinking and make some relevant suggestions or conclusions.”
Reply: We improved the critical interpretation of the previous studies presented and offered more precisely our critical opinion on the issue. For instance, the simple summary has been revised as following: “Cardiovascular (CV) risk factors and disease are increasingly reported among survivors to anticancer treatments as those treatments are effective in improving prognosis in patients affected by malignancy. It is unclear the extent to which drugs used in general population to prevent ischemic heart disease, valvular heart disease, heart failure, and aortic syndromes, which are related to CV risk factors and preclinical CV disease, are effective in survivors to anticancer treatments. Among those who survived to anticancer treatments, prognosis after cardiac surgery treatments may differ from that in the general population, and may require specific pre-surgery risk assessment.” The conclusions of the abstract have been revised as following: “Conclusions: it is unclear whether interventions to dominate cancer- and anticancer treatments related metabolic syndromes, chronic inflammation and endothelial dysfunction, leading to IHD, nonIHD, VHD, HF, and AoS, are effective in survivors to anticancer treatments as in the general population. When CV diseases require cardiac surgery, survivors to anticancer treatments may be a population at specifically elevated risk rather than affected by a specific risk factor.”
General conclusions have been revised as following “Current anticancer therapies are associated with improved survival in patients fighting cancer. However, cancer, and anticancer treatments, may induce direct car-diac toxicity, chronic inflammation, vasculitis and coronary disease, metabolic syn-dromes, and endothelial dysfunction leading to atherosclerosis and atherothrombosis. IHD, nonIHD, VHD, HF and AoS are the common CV phenotypes of CV disease in survivors to anticancer treatments. Specific and systematic experience are lacking on the extent to which current medications used in the general population to prevent and treat CV disease, protect from the decline in LV EF and prevent HF, are effective in subjects survived to anticancer treatments. Yet, general agreement exists on the fact that CV preventions should be set on aggressive targets in survivors to anticancer treatments, which intrinsically consider such a group of patients with a specifically elevated risk of events rather than patients with a specific additional risk factor. Moreover, CV diseases, HF and AoS among survivors to anticancer treatments may require programmed cardiac surgery. However, it remains to be evaluated in deep whether previous anticancer treatment should be an indicator of risk or identify a population at specifically elevated risk when undergoing cardiac surgery. In fact, historically, anticancer therapy may have been act as a negative bias in clinically ori-ented series, registries and spontaneous observations exploring prognosis post-cardiac surgery. However, as prognosis in cancer diseases improves by specific treatments, there is an increased need to develop specific models predicting mortality, morbidity and quality of life in the mid-and long term in survivors to anti-cancer treatments, in order to establish whether cardiac surgery, including LVAD, TAH and heart trans-plantation, is a sustainable therapeutic option in such a group of patients..”
Reviewer: “Minor concerns: The word "evidence" is always uncountable."treatmetns" should be treatments”
Reply: The sentence in the abstract has been revised as following “Substantial data lack on whether current….”. Manuscript title has been revised as following: “Cardiotoxicity, Cardioprotection, and Prognosis in Survivors to Anticancer Treatment undergoing Cardiac Surgery: unmet needs” We apologize for the typo, which has been amended.

Reviewer 3 Report
1. The article says nothing about anthracyclines. Anthracyclines are part of modern treatment regimens, including Brest Cancer. This is type 1 cardiotoxicity, and irreversible. The drugs cause the greatest number of complications, primarily systolic dysfunction and heart failure. A section on AС should be added.
2. Trastuzumab is most often used in combination with AС. Therefore, cardiotoxic effects are superimposed on both drugs. This should be emphasized.
Author Response
Reviewer 3
We would like to thank the Reviewer very much for reading our manuscript very carefully and providing comments found very useful to revise it. With regard to the point raised:
Reviewer: “The article says nothing about anthracyclines. Anthracyclines are part of modern treatment regimens, including Brest Cancer. This is type 1 cardiotoxicity, and irreversible. The drugs cause the greatest number of complications, primarily systolic dysfunction and heart failure. A section on AС should be added.”
Reply: In the Table 1, anthracyclines were been reported as agents associated with the phenotype of nonischemic heart disease. Aspecific section on anthracyclines has been added, and underlined that such a treatment it is the one most likely leading to asymptomatic ventricular dysfunction or overt heart failure: paragraph 3.1.1: “Anthracycline antineoplastic agents, such as doxorubicin, are effective component and widely used in adjuvant chemotherapy against breast cancer, lymphomas, leuke-mias, and sarcomas. The model of directly induced myocardial apoptosis has been described with anthracyclines as a predominantly non-inflammatory non-atherothrombotic myocardial damage predisposing to HF. Myocardial damage caused by anthracyclines have been documented by several experimental studies, and recognized post-marketing clinical practice. Non-IHD characterized by systolic dys-function may progress to overt HF and untoward nonfatal and fatal events [25]. Anthracycline-related cardiotoxicity is cumulative-dose-dependent. It is well-established that oxidative stress contributes to cardiomyocyte dysfunction and death related to the use of anthracyclines. Mitochondrial dysfunction is responsible for the large part of the oxidative stress as documented previously [26]. Careful and aggressive monitoring of cardiac function is needed in patients treated with anthracyclines to prevent major CV events [27]. In a study conducted in 120 patients, cardiac function recovery after anthracycline-induced LVEF reduction was reported in less than 10% of the treated [28]. LV EF recovery, partial of up to nor-mal, may increase significantly by the use of angiotensin I converting enzyme inhibi-tors and/or carvedilol [29], especially when introduced within 6 months from the an-ticancer treatments [30].”
Reviewer: “Trastuzumab is most often used in combination with AС. Therefore, cardiotoxic effects are superimposed on both drugs. This should be emphasized.
Reply: We appreciated very much the comment. Indeed, we proposed Table 1 where the potential impact of anticancer treatments is reported by phenotypes, aiming at emphasizing the untoward synergistic effects that combinations of those anticancer treatments may determine. Nevertheless, we agree on emphasizingspecifically that trastuzumab is used often in combination with anthracyclines, which predisposes to a combination of untoward effects on myocardial function and likelihood to develop ventricular dysfunction and over heart failure. (Paragraph 3.1.2, last sub-paragraph: “Trastuzumab is most often used in combination with AС. The cardiotoxic effects are superimposed on both drugs. It was showed that the prevalence of New York Heart Association classification (NYHA) grade III or IV cardiac dysfunction may reach 16% among persons receiving the two medications in combination including also cyclo-phosphamide. Of those, the 8% developed symptomatic cardiac dysfunction and withheld trastuzumab[40], impacting unfavorably the cancer-related prognosis.”)
Round 2
Reviewer 2 Report
-